# Cryo-EM structure of the hibernating *Thermus thermophilus* 100S ribosome reveals a protein-mediated dimerization mechanism

Rasmus Kock Flygaard [1], Niels Boegholm [1], Marat Yusupov[2,3] & Lasse B. Jenner [2]

In response to cellular stresses bacteria conserve energy by dimerization of ribosomes into inactive hibernating 100S ribosome particles. Ribosome dimerization in *Thermus thermophilus* is facilitated by hibernation-promoting factor (*Tt*HPF). In this study we demonstrate high sensitivity of *Tt*100S formation to the levels of *Tt*HPF and show that a 1:1 ratio leads to optimal dimerization. We report structures of the *T. thermophilus* 100S ribosome determined by cryo-electron microscopy to average resolutions of 4.13 Å and 4.57 Å. In addition, we present a 3.28 Å high-resolution cryo-EM reconstruction of a 70S ribosome from a hibernating ribosome dimer and reveal a role for the linker region connecting the *Tt*HPF N- and C-terminal domains in translation inhibition by preventing Shine−Dalgarno duplex formation. Our work demonstrates that species-specific differences in the dimerization interface govern the overall conformation of the 100S ribosome particle and that for *Thermus thermophilus* no ribosome-ribosome interactions are involved in the interface.

[1] Department of Molecular Biology and Genetics, Aarhus University, 8000 Aarhus C, Denmark. [2] Department of Integrated Structural Biology, Institute of Genetics and Molecular and Cellular Biology, CNRS UMR710, INSERM U964, University of Strasbourg, Strasbourg 67000, France. [3] Institute of Fundamental Medicine and Biology, Kazan Federal University, Kazan 420008, Russia. Correspondence and requests for materials should be addressed to L.B.J. (email: lasse@igbmc.fr)

Synthesis of proteins[1] is one of the most energy demanding cellular processes[2,3] calling for bacterial cells to regulate this tightly. The ability to perform such regulation becomes even more critical for bacterial proliferation in times of growth challenging cellular stresses, e.g., antibiotic exposure, nutrient starvation, or entry into stationary phase[4]. In bacteria, one such regulatory adaptation is to decrease ribosomal activity by formation of translationally inactive 100S ribosome dimers[5–7].

Ribosome dimerization has been most extensively studied in the gamma proteobacterium *Escherichia coli* where two proteins, ribosome modulation factor (RMF)[5], and hibernation-promoting factor (HPF)[8,9], are required for 100S ribosome formation. During exponential growth, 100S ribosomes are not observed in *E. coli* but in the stationary phase they are formed coincident with expression of RMF, and ribosome dimerization was shown to be dependent on RMF[6,10]. Binding of HPF to 70S ribosomes does not induce ribosome dimerization. However, binding of RMF lead to formation of 90S dimers that can then be further stabilized and transformed into 100S dimers by HPF binding[7,9]. Structural models from an X-ray crystallographic study of chimeric complexes of the *T. thermophilus* 70S ribosome with *E. coli* RMF and HPF proteins show both proteins to bind at specific sites on the small subunit[11]. RMF binds close to the 3′ end of 16S rRNA interacting with nucleotides upstream of the anti-Shine–Dalgarno (anti-SD) sequence whereas HPF binds at a position overlapping with binding sites for A, P, and E-site tRNAs[11].

Formation of 100S ribosome dimers has also been observed in non-gamma proteobacteria[12]. Common for those bacteria is the lack of an RMF homologous protein and the presence of a long HPF homologous protein (LHPF)[7]. Long HPF proteins share a conserved N-terminal domain (NTD) that is homologous to the short HPF protein of *E. coli* but the appended C-terminal domain (CTD) bears no homology to the RMF protein[13]. In *Staphylococcus aureus*, 100S ribosome formation is dependent on a long HPF protein called *Sa*HPF[13] and 100S ribosomes can be observed in exponential growth phases as well as stationary phases[12–14]. The ability of *S. aureus* to form 100S ribosomes is critical for long-term viability and for suppression of translation of specific mRNAs[14]. Similarly, in *Listeria monocytogenes* and *Bacillus subtilis*, formation of 100S ribosomes has been shown critical for regrowth and survival in competitive cultures[15,16] and for *L. monocytogenes* ribosome hibernation has been shown to mediate tolerance to certain antibiotics[17]. Biophysical experiments[12] together with structural studies, employing negatively stained samples, cryo-electron microscopy (cryo-EM) combined with single particle analysis, and cryo-electron tomography (cryo-ET) have revealed that 100S ribosome formation is quite different depending on whether it occurs in a gamma proteobacteria having a short HPF and RMF or whether the 100S is from a non-gamma proteobacteria having a long HPF.

In gamma proteobacteria such as *E. coli*, the two 70S ribosomes in the dimer interact through their 30S subunits in an end-on fashion[18–20] where binding of RMF and HPF proteins promote a conformational change within the small subunit, making ribosomal protein uS3 interact with uS2 in the paired ribosome[20]. In the non-gamma proteobacteria *B. subtilis*, *S. aureus*, and *Lactococcus lactis*, the two 70S ribosomes are more staggered with respect to each other. They still interact trough the small subunit[21–25] but display higher stabilities compared with ribosome dimers from *E. coli*[12]. The NTD of LHPF was observed to overlap with tRNA binding sites, and the CTD of LHPF forms a homodimer with the CTD of the LHPF bound to the other ribosome copy in the dimerization interface. The LHPF-CTD homodimer interacts with uS2, bS18, and h40 of the 16S rRNA, stabilizing the 100S ribosome dimer[22–25]. Further stabilization of the

dimerization interface arises from 16S rRNA h26 of one ribosome copy interacting with the uS2 protein on the other ribosome copy (see Supplementary Figure 1A, B). The linker between the LHPF NTD and CTD in published 100S reconstructions could not be clearly modeled, but it has been suggested to extend toward a region close to the 3′ end of the 16S rRNA[23,24].

Studies on formation of 100S ribosomes in *T. thermophilus* have shown that similar to *S. aureus*, 100S ribosomes are present throughout all growth phases and 100S ribosome formation is dependent on a long HPF protein, *Tt*HPF[12,14]. In vitro studies also indicate that for both *T. thermophilus* and *S. aureus* excess molar ratios of the long HPF protein inhibits 100S ribosome formation[12,14]. However, when comparing 100S ribosome models derived from cryo-EM reconstructions to crystal structures of the *T. thermophilus* ribosome, it is immediately clear that whereas the length of h40 in *T. thermophilus* ribosome might still allow for interactions with *Tt*HPF-CTD, the shorter h26 of the 16S rRNA is unlikely to mediate interactions with uS2 and *Tt*HPF-CTD (Supplementary Fig. 1C–H) unless the *T. thermophilus* 100S ribosome dimer is in a different conformation. This prompted us to investigate how the ribosome dimerization interface in *T. thermophilus* is stabilized.

In this study we report cryo-EM reconstructions of in vitro formed *T. thermophilus* 100S (*Tt*100S) ribosomes revealing a ribosome dimerization interface that unlike what has been observed before rely solely on interactions between the homodimer of *Tt*HPF-CTD and ribosomal protein uS2. Using biophysical experiments on in vitro formation of *Tt*100S we show that *Tt*100S ribosome formation is dependent on *Tt*HPF and that the formation is highly sensitive to the molar ratio of the *Tt*HPF protein. Furthermore, the improved quality of our electron density allowed us to confidently build an atomic model of *Tt*HPF and analyze the *Tt*HPF-NTD interactions to the ribosome in the highest detail compared with the previous structures from *B. subtilis*, *S. aureus,* and *L. lactis*.

## Results

**Formation of 100S ribosomes by *Tt*HPF**. Purified *Tt*HPF protein without ribosome present exists as a homodimer in solution as evident from the elution volume in size exclusion gel filtration analysis (Supplementary Fig. 2A). The early elution of *Tt*HPF compared to standard proteins, also suggests that a TtHPF homodimer does not adopt a globular shape but rather a more extended conformation. This is in agreement with observations of other long HPF proteins[23–25]. This homodimeric *Tt*HPF protein retains its expected biological activity to induce formation of 100S ribosomes (Supplementary Fig. 2B).

We investigated the in vitro dependency of *Tt*100S ribosome formation on *Tt*HPF by analytical ultracentrifugation (AUC) on isolated *Tt*70S ribosome mixed with purified *Tt*HPF in a series of molar ratios assuming one copy of *Tt*HPF to bind one *Tt*70S ribosome (Fig. 1a and Supplementary Fig. 3). *Tt*HPF-mediated formation of 100S ribosomes was found to be maximal at equimolar ratios of *Tt*HPF to 70S ribosome (Fig. 1a, b). Even at sub-molar ratios, formation of 100S ribosome was observed along with observations of strong inhibition of 100S formation at molar excesses of *Tt*HPF to 70S ribosome (Fig. 1b and Supplementary Fig. 3). These observations of a sharp transition in promotion or inhibition of 100S formation by *Tt*HPF agrees with previous studies on 100S ribosome formation in both *T. thermophilus* and *S. aureus*[12,14]. However, the results presented here show a much stronger inhibitory effect of *Tt*HPF on *Tt*100S ribosome formation in vitro at even moderate molar excess. This observation could be explained by a strong binding of *Tt*HPF to the *Tt*70S ribosome, whereby *Tt*HPF binding sites on

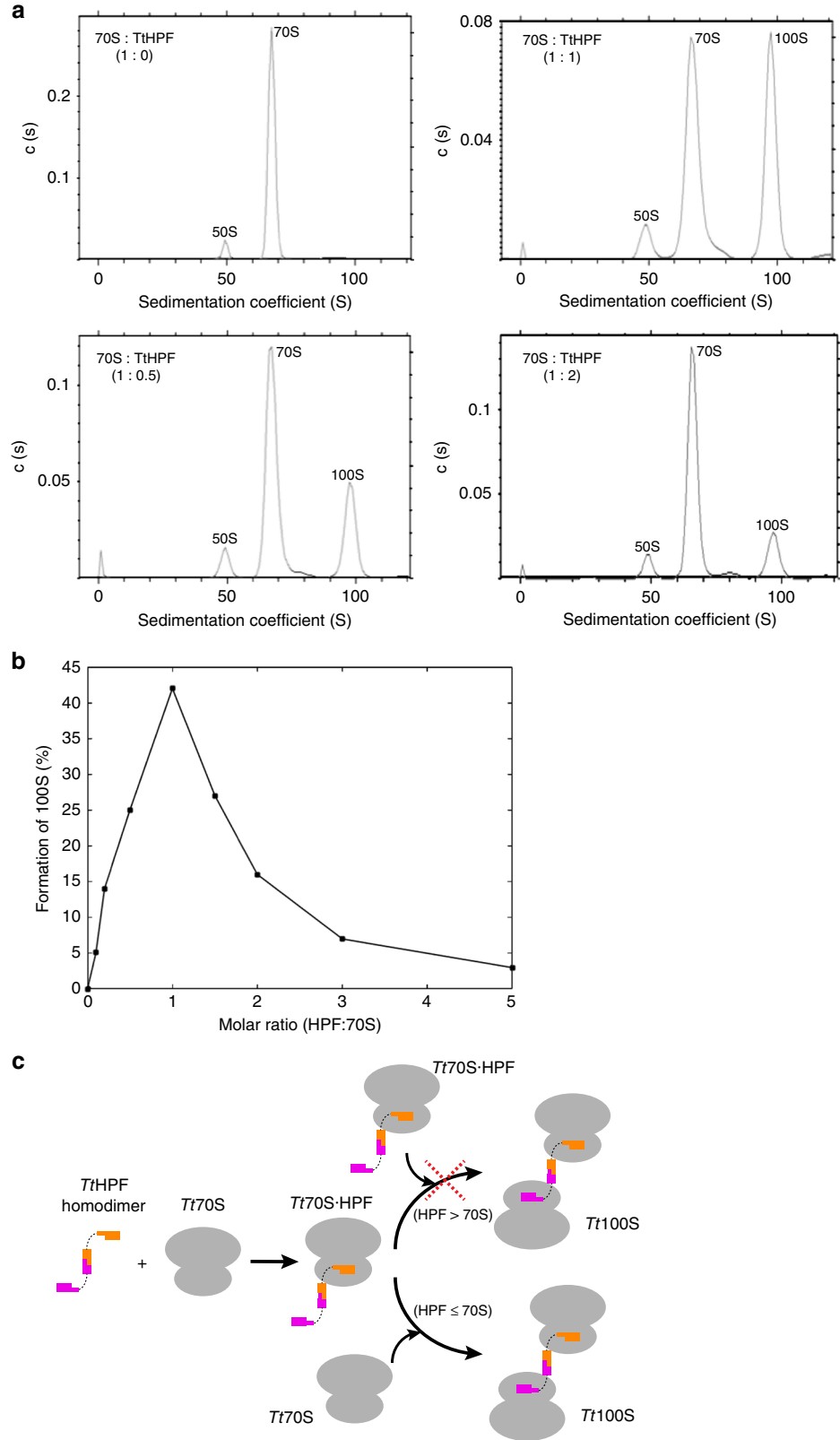

ribosomes are saturated and thus acting inhibiting to Tt100S ribosome formation (Fig. 1c). We did not observe a complete conversion of 70S to 100S dimers which lead us to believe that either TtHPF is not fully active at the temperature where the experiment was performed or perhaps a population of the 70S

ribosomes are protected from dimerization in some way that we could not detect. Given the AUC results on 100S formation, an equimolar ratio of purified TtHPF and 70S ribosome was used in 100S formation for cryo-EM experiments (Supplementary Fig. 2C).

**Fig. 1** Analysis of in vitro *Tt*HPF dependent formation of *Tt*100S. **a** Analytical ultracentrifugation sedimentation profiles show 70S ribosome as control (upper left), 70S ribosome mixed with *Tt*HPF in 0.5 times molar ratio (lower left), 70S ribosome mixed with *Tt*HPF in equimolar ratio (upper right) and 70S ribosome mixed with *Tt*HPF in two times molar ratio (lower right). Formation of *Tt*100S ribosome is evident by the peak at a sedimentation coefficient of 100S. **b** Graphical representation of *Tt*100S ribosome formation from all AUC experiments. Formation of *Tt*100S ribosome by *Tt*HPF is maximal in the case where the molar ratio of *Tt*HPF to *Tt*70S is 1:1. See also Supplementary Figure 3. **c** Schematic illustration of *Tt*HPF and *Tt*70S binding events leading to *Tt*100S ribosome formation. Binding of one NTD of *Tt*HPF homodimer to *Tt*70S leads to a complex of *Tt*70S·*Tt*HPF. In the case of sub- or equimolar ratios of *Tt*HPF and *Tt*70S, binding of a vacant *Tt*70S ribosome to the free NTD of the *Tt*70S·*Tt*HPF complex leads to *Tt*100S formation. However, in the case of *Tt*HPF being present in excess molar ratios, *Tt*100S ribosome formation is inhibited because all *Tt*70S ribosomes bind a TtHPF homodimer

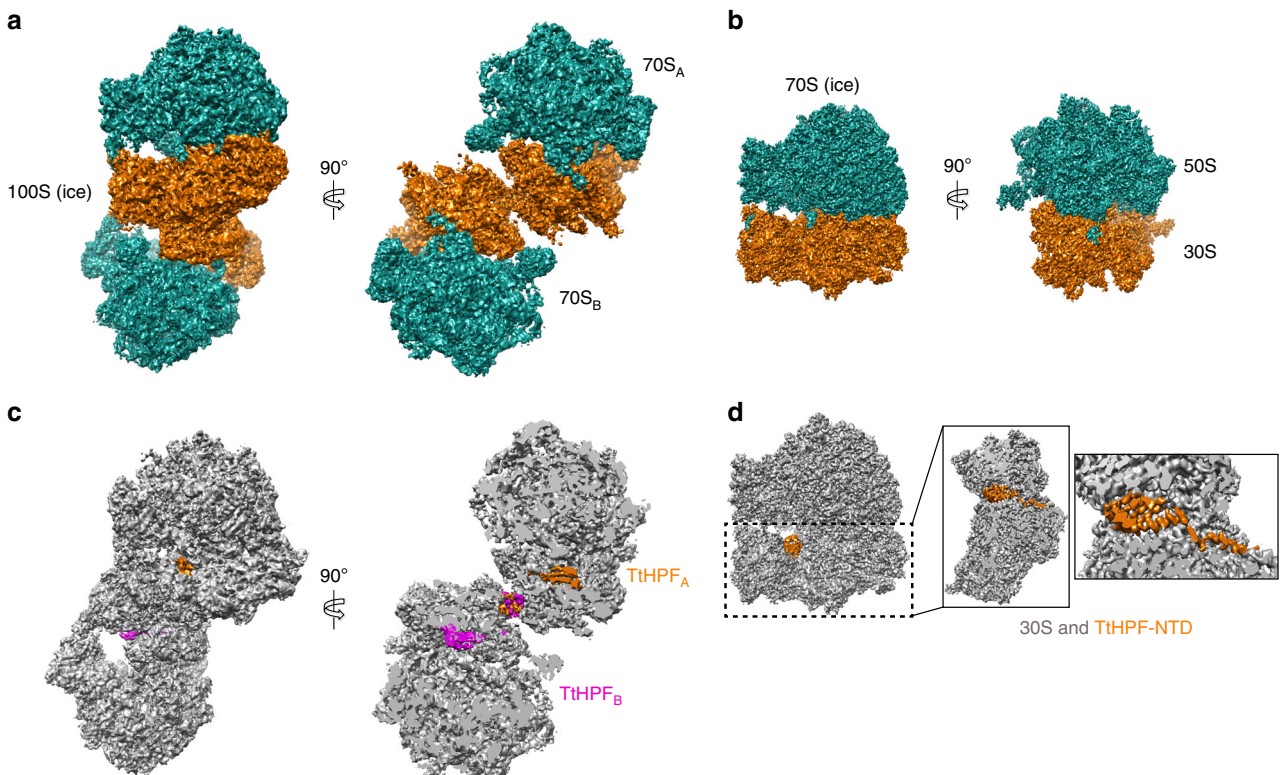

**Fig. 2** Cryo-EM structures of 100S (ice) and 70S (ice). **a** Orthogonal views of 100S (ice) with 50S subunits in green and 30S subunits in orange showing the two 70S copies constituting the 100S particle. **b** Orthogonal views of 70S (ice), coloring of subunits as in (**a**). **c** Views of 100S (ice) and slice-through view with both 70S ribosome copies colored in gray and the two *Tt*HPF protein molecules colored in orange and magenta showing location of *Tt*HPF-NTD and CTD within the 100S ribosome dimer. **d** View of 70S (ice) with *Tt*HPF-NTD colored in orange. Close-up views on 30S subunit show location of *Tt*HPF-NTD and the linker region. There was no density for the *Tt*HPF-CTD in 70S (ice) reconstruction

**Cryo-EM structure determination of *Tt*100S ribosome**. For the first cryo-EM data set, in vitro formed and purified *Tt*100S ribosomes were spotted on unsupported cryo-grids and imaged in the vitreous ice state. Automatic particle picking did not perform to our satisfaction, thus care was taken by manually inspecting all picked positions to avoid ribosome dimers lying very close as well as to include non-picked ribosome dimer particles. During 3D classification, one class showed distinct density features for two ribosomes in a dimer particle (Supplementary Fig. 4) with an internal C2 symmetry that resulted in a 3D reconstruction with the highest average resolution reported yet for a 100S ribosome of 4.57 Å (Fig. 2a and Supplementary Fig. 4). From the same data set, 3D classification also showed a class with a single high resolved 70S ribosome with almost no density for the other ribosome copy of the dimer (Supplementary Fig. 4). The particles in this class were used in focused refinement using a 70S ribosome mask resulting in high-resolution 3D reconstruction of a 70S ribosome with an average resolution of 3.28 Å (Fig. 2b and

Supplementary Fig. 4). We refer to the two 3D reconstructions as 100S (ice) and 70S (ice), respectively. In the 70S (ice) reconstruction a high resolved density corresponding to *Tt*HPF-NTD was observed (Fig. 2b, d), however, no density was observed for the *Tt*HPF-CTD. In the 100S (ice) reconstruction densities were observed for both the *Tt*HPF-NTD and CTD bound to both ribosome copies (Fig. 2c). An atomic model of *T. thermophilus* 70S ribosome was initially fitted in the 70S (ice) reconstruction and *Tt*HPF-NTD was built using a crystallographic model as template (see Materials and Methods). In the 70S (ice) reconstruction we observed clear density for the linker region connecting the NTD and CTD, hence we modeled *Tt*HPF residues 2–122 (Fig. 2d and 3a).

**TtHPF-NTD interactions with 30S subunit**. *Tt*HPF-NTD binds to the 30S subunit at a position between the head and body as previously observed in 100S ribosomes from other species[22–25] as

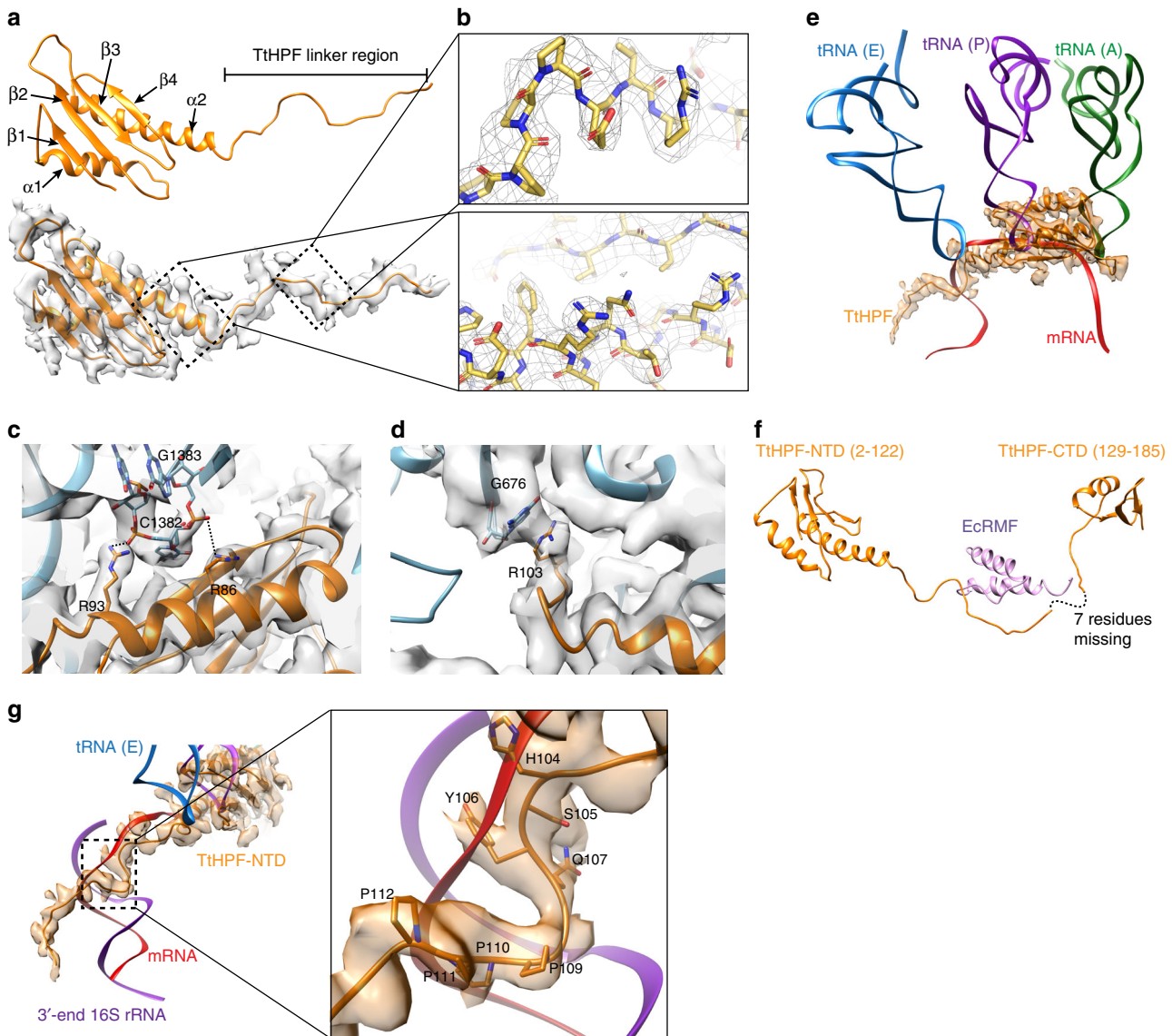

**Fig. 3** Structure of *Tt*HPF-NTD and its interactions with the 30S subunit. **a** Structure of *Tt*HPF-NTD shown in orange cartoon. The corresponding cryo-EM density of 70S (ice), shown as semi-transparent gray surface, shows clear density for linker region. Secondary structure elements are labeled. **b** Close-up view on the high quality density map of the *Tt*HPF-NTD (upper) and linker (lower) with model inside. Side chains are clearly resolved in the high-resolution density. **c** Close-up view on electrostatic interactions between *Tt*HPF-NTD and ribosome centered around Arg86 and Arg93 interacting with phosphate backbone on nucleotides C1382 and G1383 in h44. **d** Example of stacking interaction between *Tt*HPF-NTD residue Arg103 and nucleotide G676 of h23 in 16S rRNA. **e** Structure of *Tt*HPF-NTD and its density (orange) superimposed with A-, P-, and E-site tRNAs (green, purple, blue) and mRNA (red) bound in 70S ribosome (PDB entry 4V6F). The binding position of *Tt*HPF-NTD on the 30S subunit clearly overlaps with binding sites for all three tRNAs as well as mRNA. **f** Structure of *Tt*HPF with the missing seven residues of the linker indicated by dashed line. The structure is superimposed with the chimeric structure of *Tt*70S-RMF (PDB entry 4V8G[11]) showing the location of the *E. coli* RMF protein (violet) closely matching the binding position of the *Tt*HPF linker region. **g** Coloring as in E now with 3′ end of 16S rRNA shown in violet. The linker region of TtHPF occupies a binding position on the 30S subunit that overlaps with the helix formed by mRNA and 3′ end of 16S rRNA. Close-up view shows *Tt*HPF linker region residues His104, Ser105, Tyr106, and Gln107 overlapping with Shine–Dalgarno duplex between mRNA and 3′ end of 16S rRNA. Proline residues 109–112 overlap with mRNA binding position as well

well as for the *E. coli* short HPF[11]. Similar to other long HPF proteins, *Tt*HPF-NTD adopts a β1-α1-β2-β3-β4-α2 topology with strands β1-β2 being parallel and β2-β3-β4 anti-parallel (Fig. 3a), with high sequence conservation to other long HPF proteins (Supplementary Figure 8). Hydrophobic residues in α1, β1, and β2 as well as α2-β4 form a hydrophobic core that stabilizes the fold of *Tt*HPF-NTD (Supplementary Fig. 6A). Interactions between aromatic residues Tyr20 and the highly conserved Tyr77 (Supplementary Figure 8) further contribute to the stabilization of the *Tt*HPF-NTD. Our high quality 70S (ice) density map also

showed clear densities for side chains on *Tt*HPF-NTD residues enabling a detailed analysis of interactions to the ribosome (Fig. 3b).

The clear density observed for side chains on *Tt*HPF-NTD showed interactions predominantly with 16S rRNA through electrostatic interactions between basic residues and phosphate groups of the rRNA backbone in h30, h31, and h44, e.g., between Arg86 and Arg93 phosphates on nucleotides C1382 and G1383 in the base of h44 (Fig. 3c and Supplementary Fig. 6B–D). We also observed clear side chain density for interactions between *Tt*HPF-

NTD and 16S rRNA through stacking interactions between Tyr31 and A773 in the loop of h24 as well as by Arg103 with G676 in the loop of h23 (Fig. 3d and Supplementary Fig. 6E). The only interaction observed between the TtHPF in the 70S (ice) reconstruction and a ribosomal protein is the contact formed between Tyr25 in uS11 and Pro112 of the linker between NTD and CTD (Supplementary Fig. 6F).

The binding site of TtHPF-NTD on the 30S subunit overlaps with binding sites for tRNAs in A-, P-, and E-sites as well as the mRNA binding groove on the small ribosomal subunit (Fig. 3e). The tight tethering of TtHPF-NTD to 16S rRNA helices positions the folded NTD right in the A- and P-sites on the 30S subunit, a binding position that precludes binding of tRNA in either of these two sites (Fig. 3e). The long α2 helix and the start of the TtHPF linker region occupy a position that would cause a steric clash with tRNA in E-site (Fig. 3e). Previous studies also indicated overlap of the binding position of long HPF proteins on 30S subunit with A- and P-site tRNAs[22–25], but here we show that the TtHPF-NTD inhibitory effect on translation is due to a complete steric hindrance of tRNA binding in any of the three A-, P-, and E-sites. This also agrees with biochemical observations of reduced translation activity in in vitro assays[12,14]. In previous structures of long HPF proteins, the density for the LHPF-NTD and the linker region was poorly resolved allowing the linker region to only be traced to approximately residue 101 to 106 depending on the study[23–25]. In our 3.28 Å 70S (ice) reconstruction the electron density for both the NTD and the linker was well resolved (Figs 2d, 3a, b) and allowed us to build the TtHPF linker until residue 122, only leaving a gap of seven residues between the N and C-terminal domains (Fig. 3f). The TtHPF linker region extends toward the mRNA exit site in proximity of the 16S rRNA 3′ end (Fig. 3e), a position close to that occupied by E. coli RMF protein on the small subunit[11] (Fig. 3f). The part of the linker region closest to the 3′ end of 16S rRNA is a stretch of residues from His104 to Pro112, where the four proline residues 109–112 are arranged in a distorted poly-proline-II helix (Fig. 3g). This type of secondary structure is known for its ability to interact with nucleic acids[26]. This binding position of the TtHPF linker is in the same region where E. coli RMF[11] was observed to bind and will interfere with helix formation between Shine–Dalgarno (SD) and anti-Shine–Dalgarno (aSD) sequence (Fig. 3g) during translation initiation[27,28], hence causing inhibition of translation.

**TtHPF-CTD bridges interactions in 100S ribosome interface.**
The 100S (ice) reconstruction showed a density located in the dimerization interface of the ribosome copies (Fig. 2c) that we attributed to the TtHPF-CTD homodimer. Given only a medium resolution in this part of the reconstruction (Supplementary Fig. 4B) we collected a second cryo-EM data set this time with isolated 100S ribosome spotted on cryo-grids with a continuous amorphous carbon support. Processing of single particles essentially followed the steps of the first data set processing (Supplementary Figure 5) with 3D classification showing one class of 100S particles that aligned with C2 symmetry resulting in a final 3D reconstruction with an average resolution of 4.13 Å surpassing that of previous 100S ribosome reconstructions including our 100S (ice) reconstruction (Fig. 4a and Supplementary Figure 5). This 3D reconstruction we refer to as 100S (amc). As the conformation of TtHPF was found to be identical in the three different reconstructions (Supplementary Figure 5C), we initially thought of combining 100S particles from the two data sets (ice and amc) aiming for a higher resolved reconstruction allowing a closer analysis of the 100S dimerization interface around the TtHPF-CTD homodimer. However, although the overall conformation of the 100S (ice) and 100S

(amc) was identical, there was a slight difference when looking specifically at the dimerization interface where the 100S (amc) has two additional sites of interaction between the small subunit head domains centered at ribosomal proteins uS7 and uS9 (Supplementary Figure 7A). In addition, we observed a different conformation of H69 of 23S rRNA between 100S (ice) and 100 (amc) (Supplementary Figure 7B). Thus, we kept particles of the 100S (ice) and 100S (amc) reconstructions separate. The density around the TtHPF-CTD homodimer region in the 100S (amc) was slightly better resolved compared to the 100S (ice) reconstruction (Supplementary Figure 5B), thus TtHPF-CTD was modeled in the 100S (amc) density (Figs 4a, b). The homodimer of TtHPF-CTD adopts a conformation similar to that observed for BsHPF[23], SaHPF[24,25] and LlHPF[22] with each copy of the TtHPF-CTD forming a central three-stranded beta-sheet flanked by an alpha helix (Fig. 4a, small insert). This is in line with the high degree of sequence conservation in the CTD (Supplementary Figure 8). The central beta-sheets are extended by one strand by interacting with the other copy of TtHPF-CTD in the region connecting to the linker region (Fig. 4b). The interactions between the two beta-sheets of the two TtHPF-CTD copies are dominated by hydrophobic interactions, e.g., Ile169 on one copy interacts with Ile169 on the other copy together capping the CTD dimerization interface lined by Val160 and Val171 on both copies (Fig. 4b). Further stabilization of the CTD dimerization interface comes from intricate stacking interactions of aromatic residues Phe158 and Tyr173 tethering the two TtHPF-CTD together (Fig. 4b). This very tight interaction between the TtHPF-CTD copies resembles that observed in SaHPF-CTD[24].

Interestingly, in our Tt100S ribosome we do not observe the network of interactions in the dimerization interface observed for 100S ribosomes from other species[22–25]. As speculated from analysis of 16S rRNA secondary structure diagrams (Supplementary Figure 1), the inter-ribosome interaction between 16S rRNA h26 of one ribosome copy and uS2 on the other ribosome copy is not present in our Tt100S ribosome reconstructions from 100S (amc) nor 100S (ice) (Fig. 4c and Supplementary Figure 7C). The h26 does indeed forms a helix protruding from the small subunit, however the length of h26 is much too short to make it all the way to uS2 protein on the other ribosome to interact (Fig. 4c). Thus, no additional stabilization of the ribosome dimerization interface can be attributed to the h26-uS2 interaction.

We also do not observe an interaction similar to that between the SaHPF-CTD homodimer and 16S rRNA h40 in our Tt100S ribosome (Fig. 4d and Supplementary Figure 7D). As the length of h40 is similar to that in other species (Supplementary Figure 1H), where it makes contact with the TtHPF-CTD homodimer, the conformation of the Tt100S ribosome must be slightly different from that of 100S ribosomes from other species given there is no interaction. Thus contrary to what has been observed for other species of 100S ribosome dimers, the ribosome dimerization interface for the Thermus thermophilus 100S ribosome is centered on the TtHPF-CTD homodimer and only comprises interactions between ribosomal protein uS2 and TtHPF-CTD on the same ribosome copy with the opposite TtHPF-CTD and protein uS2. These results provide evidence that formation of Thermus thermophilus 100S ribosome dimers by long HPF proteins is attributed to the LHPF protein alone and not inter-ribosome interactions.

## Discussion
Formation of 100S ribosomes is a ubiquitous bacterial response to cellular stresses and unfavorable growth conditions[12]. In such circumstances, bacterial cells rely on second messenger signaling molecules, e.g., (p)ppGpp, to tune the process of translation to

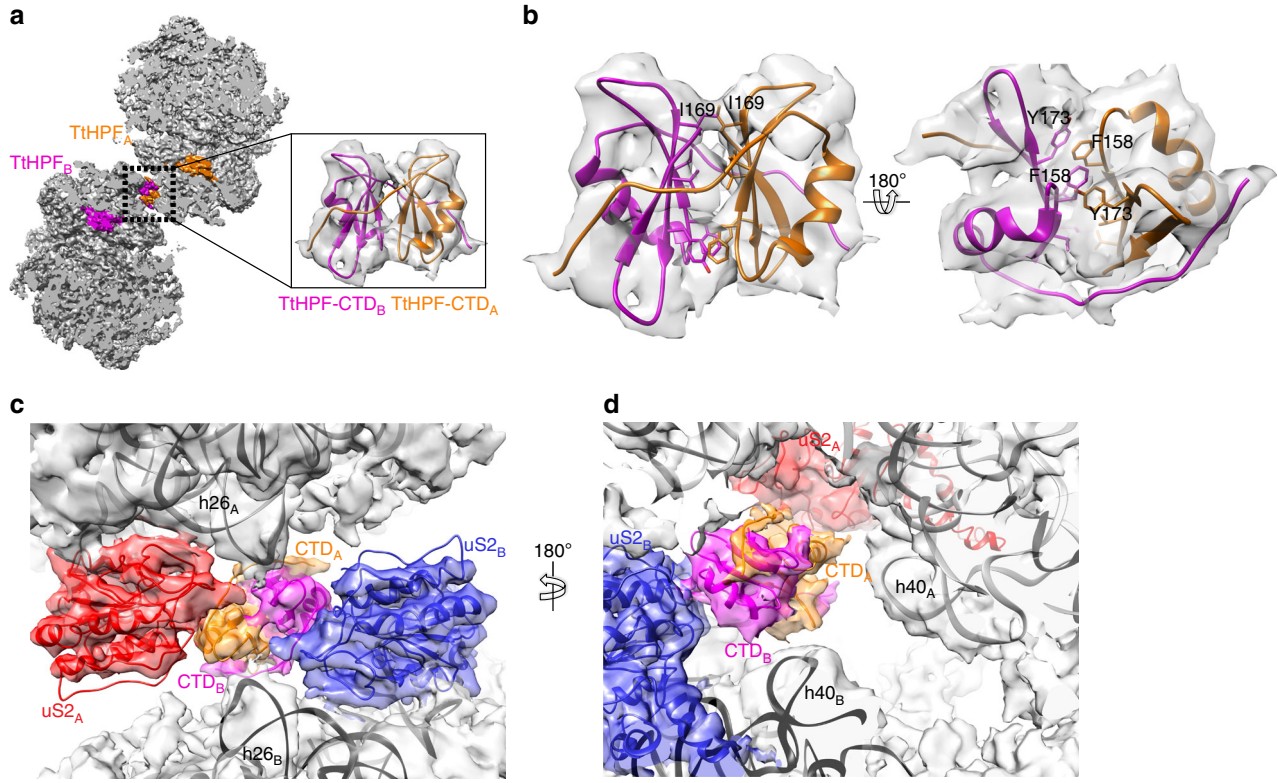

**Fig. 4** Cryo-EM structure of 100S (amc). **a** Slice-through view of 100S (amc) reconstruction with both 70S ribosome copies colored in gray and the two *Tt*HPF protein molecules colored in orange and magenta showing location of *Tt*HPF-NTD and CTD within the 100S ribosome dimer. Close-up view shows the homodimer *Tt*HPF-CTD model in the corresponding EM density. **b** Interactions between the two copies of *Tt*HPF-CTD are governed by hydrophobic interactions (e.g., by Ile169) between the two beta-sheets (left side view) and by stacking interactions of aromatic residues (right side view). **c** *Tt*100S ribosome (light gray semi-transparent surface) dimerization interface around *Tt*HPF-CTD (orange and magenta) with uS2 (red and blue models and densities) and h26 of 16S rRNA (labeled model inside 100S (amc) density). The length of h26 is too short for making interactions with uS2 on the other ribosome copy. **d** Coloring is as in C. h40 16S rRNA model is shown. There is no interaction from *Tt*HPF-CTD homodimer to either of the h40 on the ribosomes

conserve energy[4]. Though the mechanism of ribosome dimerization is different between gamma proteobacteria and non-gamma proteobacteria[29], the regulation of the proteins involved in ribosome dimerization seems conserved. In *E. coli*, the protein responsible for ribosome dimerization, RMF[6], is transcriptionally upregulated in response to secondary messengers (p)ppGpp and cAMP during nutrient starvation[30,31]. Similarly, in organisms employing LHPF proteins for ribosome dimerization their regulation is also dependent on (p)ppGpp[32,33]. The stringent response is also conserved in *T. thermophilus*, thus making it likely that *Tt*HPF is regulated by (p)ppGpp in response to cellular stresses[34].

In this study we presented the cryo-EM reconstruction of *Tt*100S ribosome along with biophysical characterization of 100S formation as a function of *Tt*HPF molar ratio. The results obtained from AUC analysis showed increasing formation of 100S ribosome in response to increasing molar ratios of *Tt*HPF with a maximum conversion of 70S ribosome to 100S ribosome dimers at equimolar ratios of *Tt*HPF (Fig. 1). At excess molar ratios of *Tt*HPF to ribosome, we observed a strong inhibition of 100S ribosome formation (Supplementary Fig. 3) in agreement with previous experiment[12,14], but with our results showing a much higher sensitivity of *Tt*100S formation to *Tt*HPF amounts. Thus, similar binding events likely govern formation of 100S ribosome dimers in *T. thermophilus* and *S. aureus*[14]. Despite the clear formation of *Tt*100S ribosomes indicated by the 100S peaks

in the sedimentation profiles, we also observed a 70S sedimentation peak that we naturally attribute to 70S ribosomes. In the experiment with *Tt*70S ribosome and *Tt*HPF mixed in 1:1 molar ratio, approximately only half of the *Tt*70s ribosomes are converted to *Tt*100S ribosomes. Whether this is a reflection of our purified *Tt*HPF protein not being fully active is difficult to assess. It could also be that the temperature of 20 °C during the AUC experiment is inhibiting for 100S ribosome formation given the thermophilic nature of the source organism. However, no other experiments of 100S formation from other species has ever reported complete conversion—often the conversion rate has been quite low[12,14,24]. Perhaps the incomplete conversion to 100S reflects yet undiscovered properties of the mechanism of dimerization or that a certain populace within the ribosome pool are protected from dimerization. This would prevent a complete shutdown of translation and ascertain that some 70S ribosomes are still available for protein production during stress. The decrease in *Tt*100S ribosome formation upon excess molar ratios of *Tt*HPF we interpret as all possible binding sites on single ribosomes filling up with *Tt*HPF homodimers effectively leaving no vacant *Tt*70S ribosomes to form 100S dimers (Fig. 1c).

In the crystal structure of the chimeric complex of *T. thermophilus* 70S ribosome with *E. coli* HPF and RMF reported by Polikanov et al.[11], a model for the 100S ribosome was proposed based on an *E. coli* 100S reconstruction[19]. It was suggested that the 30S subunits in the 100S ribosome would form two points of

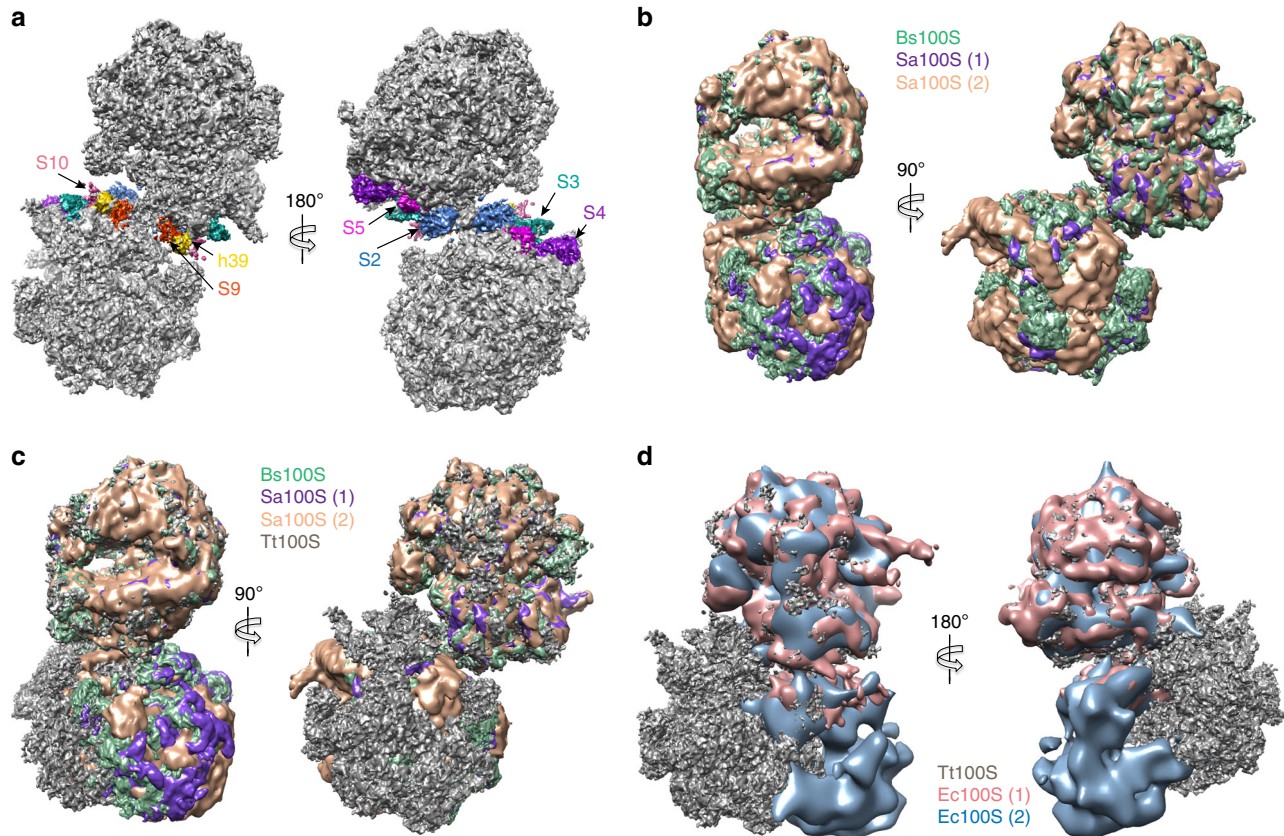

**Fig. 5** Conformation of *Tt*100S compared with other 100S structures. **a** Location of interacting proteins and rRNA as suggested by Polikanov et al. are shown in colors, uS2 in blue, uS3 in green, uS4 in purple, uS5 in magenta, uS9 in orange, uS10 in salmon, and h39 in gold. *Tt*100S ribosome density is colored gray. Clearly, the proteins and h39 are located far from each other. **b** *B. subtilis* 100S (EMD-3664, green) and *S. aureus* 100S (EMD-3637, purple and EMD-3638, salmon) in orthogonal views showing high degree of conformational conservation between the 100S molecules. **c** Same colors as in B. *Tt*100S (ice) shown in gray and superimposed to *B. subtilis* 100S. The orthogonal views clearly show are different conformation of the two ribosomes in *Tt*100S. **d** *Tt*100S (ice) in gray compared with *E. coli* 100S reconstructions (EMD-5174, pink and EMD-1750, light blue). The staggered conformation of *Tt*100S is very different from the *E. coli* 100S conformation, which is more back-to-back for the 30S subunits

contact centered at ribosomal proteins uS9, uS10, and h39 of 16S rRNA and around ribosomal proteins uS2, uS3, uS4, and uS5[11]. However, our structures clearly show this not to be the case. In our *Tt*100S ribosome reconstructions, the ribosomal proteins proposed to interact are located far from each other on the respective ribosomes within the dimer (Fig. 5a).

So far structures of LHPF-mediated 100S ribosomes from *B. subtilis* and *S. aureus* determined by cryo-EM show a high degree of conformational homogeneity with the ribosomes constituting the 100S ribosome dimers superposing very well (Fig. 5b)[22–25]. A common feature for the *Bs*100S and *Sa*100S ribosomes is the LHPF homodimer protein that facilitates ribosome dimerization in all these structures. The LHPF-CTD interacts with uS2 and h40 and additional stabilization of the 100S ribosome dimerization interface comes from inter-ribosome interactions by h26 and uS2 (Fig. 5b and Supplementary Fig. 1). As shown in our structure of the *Tt*100S ribosome, the dimerization interface is not stabilized by the h26-uS2 interaction between the ribosome copies in the 100S dimer (Fig. 4e) nor by interactions between *Tt*HPF-CTD and h40 (Fig. 4f). This leaves only *Tt*HPF-CTD interaction with uS2 to stabilize the dimerization interface. Earlier speculations on *Tt*100S ribosomes adopting a tilted conformation to bring h26 into interaction distance with uS2[24] are thus not correct according to our 100S structure. We observe a slightly altered staggered conformation of the *Tt*100S ribosome, both the *Tt*100S

(ice) and *Tt*100S (amc), compared with other 100S ribosome structures (Fig. 5c and Supplementary Figure 7E). When superposing one 70S copy of our *Tt*100S ribosome to fit with a 70S copy of *Bs*100S and *Sa*100S, the superposed 70S copy fits very well. However, the other 70S copy of our *Tt*100S ribosome does not superpose well with the other ribosome copy on the *Bs*100S and *Sa*100S ribosomes indicating a different conformation for the *Tt*100S ribosome. Based on the results we have presented here for the *Tt*100S ribosome structure, the *Tt*100S ribosome dimerization follows that of other organisms relying on LHPF proteins for 100S ribosome formation[12,22–25], where 100S ribosome dimerization is a result of binding of a homodimeric LHPF protein that brings the two ribosomes into a staggered conformation. The differences that we do see, e.g., the h26-uS2 interaction, between our 100S and 100S ribosomes from other species having long HPFs are overall not very large but might reflect species-specific differentiations of the LHPF-ribosome dimerization interface to modulate or regulate stabilization according to species-specific needs.

As expected since *T. thermophilus* has a LHPF protein, the *Tt*100S ribosome is markedly different in its staggered conformation compared with the structure of *E. coli* 100S ribosome (Fig. 5d) where RMF and a short HPF are required[7]. The binding of these two proteins induces the formation of *Ec*100S ribosomes in which the 30S subunits interact in a back-to-back fashion

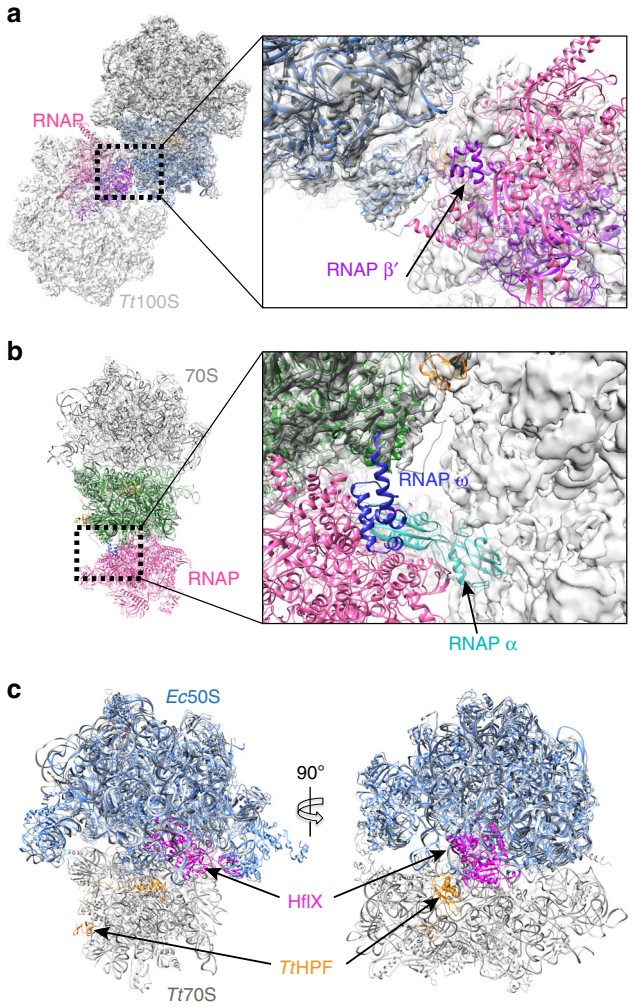

**Fig. 6** Comparison of *Tt*100S ribosome to complexes of 30S-RNAP and 50S-HflX. **a** Structure of 30S-RNAP (PDB entry 6AWB) with 30S subunit in blue, RNAP in pink and RNAP β′ subunit in purple fitted in *Tt*100S (ice) density. For clarity only one model of *Tt*70S ribosome is shown (dark gray). In this conformation of 30S-RNAP, the RNAP would be sterically prevented by the 100S ribosome dimer from occupying this binding position close to uS2 (close-up view). **b** Overlay structure of one ribosome copy of *Tt*100S with *E. coli* expressome (PDB entry 5MY1). Expressome is shown with 30S subunit in green, RNAP in pink, RNAP ω subunit in blue and RNAP α subunit in cyan. The close-up view shows that the interaction between RNAP ω subunit and uS2 is not sterically hindered by the 100S ribosome (gray semi-transparent surface), however, the RNAP α subunit clashes with the 100S ribosome close to the L1 stalk causing steric blocking of this 30S-RNAP conformation by the 100S ribosome. **c** Comparison of our *Tt*70S ribosome and *Tt*HPF model from 70S (ice) reconstruction (light gray) to the cryo-EM structure of *E. coli* 50S subunit with HflX protein bound (light blue) from PDB entry 5ADY. *Tt*HPF-NTD is shown in orange and HflX is shown in magenta. The two proteins bind at different positions enabling binding of both at the same time thus HflX would still be able to bind 100S ribosomes and disassemble them

(Fig. 5d). Whereas the NTD of *Tt*HPF occupies a binding site on the 30S subunit similar to the binding site for *E. coli* HPF[11], interfering with binding of mRNA and tRNAs (Fig. 2b), the *Tt*HPF-CTD binds at a site close to uS2 which is radically different from RMF binding site on 30S[11]. However, the mechanism proposed for RMF in inhibition of translation[11] possibly applies

for the *Tt*HPF linker region connecting the NTD and CTD. The linker region extends to a region, which would clash with formation of the SD-aSD helix during translation initiation (Fig. 3g) providing yet another inhibitory mechanism for *Tt*HPF on translation. Alternatively, the presence of the linker region in the same area as RMF is coincidental and the role of the linker region may be in synchronization of the dimerization by the CTD and the blocking of mRNA and tRNA binding by the NTD.

Although the precise role for sequestering ribosomes as translational inactive 100S ribosome dimers is not known, various reasons can be envisioned. Structural studies of the bacterial RNA polymerase (RNAP) in complex with the bacterial small ribosomal subunit have shown that RNAP interacts through its β′-subunit with ribosomal protein uS2 and h40 in a conformation where the mRNA exit tunnel of RNAP is positioned near the 3′ end of 16S rRNA[35]. If this RNAP-30S subunit structure represents an intermediate during translation initiation, then this step would be inhibited by 100S ribosome conformation as uS2 and h40 are buried within the 100S ribosome dimerization interface and thereby inaccessible[35] (Fig. 6a). In a different study showing how RNAP binds to the complete 70S ribosome, the RNAP mRNA exit tunnel faces the mRNA entry site on the ribosome between uS3, uS4, and uS5. In this conformation the ω-subunit of RNAP interacts with ribosomal protein uS2[36]. The complex of RNAP-70S ribosome referred to as "the expressome" was shown to form only during transcription elongation[36]. As in the former RNAP-30S subunit case with uS2 being embedded within the 100S ribosome dimerization interface, interactions between RNAP and 70S ribosome would be sterically hindered causing a decrease in translation activity. When overlaying the structure of the expressome with our *Tt*100S ribosome we see that it is not the RNAP ω-subunit that is sterically prevented from binding to uS2, but rather the RNAP subunit α that sterically clashes with the 100S ribosome at a position close to the L1 stalk (Fig. 6b). However, in either case, the arrangement of the 100S ribosome precludes the formation of the expressome.

The conformation of the 100S ribosome might also reflect the cellular need to disassemble and recycle the ribosomes to be able to use them in translation again. Studies have reported a universally conserved GTPase named HflX that binds the large ribosomal subunit[37]. HflX was shown to dissociate 70S ribosome into small and large subunits in a GTP-dependent manner, with the HflX protein staying attached to the large subunit after dissociation of 70S ribosome[38]. In *S. aureus*, HflX was also shown to prevent 100S ribosome formation also dependent on GTP, even in the presence of *Sa*HPF[39]. A cryo-EM structure of the *E. coli* 50S subunit with HflX bound, located the protein in the ribosomal A-site causing structural rearrangements within the 50S subunit likely to be responsible for the 70S ribosome dissociating capabilities of HflX[38]. Superposing our cryo-EM structure of *Tt*HPF bound to the ribosome with the *E. coli* 50S-HflX structure shows no steric overlap between *Tt*HPF and HflX (Fig. 6). Thus, it may be possible for HflX to bind a 100S ribosome and disassemble it.

## Methods

**Cloning and expression of *Tt*HPF**. Genomic DNA from *Thermus thermophilus* cells was extracted using Trizol (Sigma). The *Tt*HPF gene (UniProt Q5SIS0) was PCR amplified using forward primer also encoding a TEV-protease cleavage site, 5′-GACGACGACAAGATGGAAAACCTGTATTTTCAGGGCATGAACATCTA-CAAGCTCATCG-3′, and reverse primer 5′-GAGGAGAAGCCCGGTTCATCA GGCGGGCTCTATAAGGC-3′. The PCR product was cloned in pET46-Ek/LIC plasmid (Merck) and verified by sequencing. Protein was expressed in *Escherichia coli* BL21 (DE3) cells by auto-inducing ZYP-5052 growth medium[40] supplemented with 100 μg/mL ampicillin. Cells were grown at 37 °C until reaching OD$_{600}$ of 0.6 followed by overnight incubation at 18 °C. Harvested cells were resuspended in lysis buffer (50 mM Hepes/KOH pH 7.5, 300 mM KCl, 5 mM MgCl$_2$, 20 mM

imidazole, 10% v/v glycerol, 1 mM DTT) supplemented with protease-inhibitor tablet (Sigma) and 5 U/mL DNaseI (ThermoFisher).

**Purification of *Tt*HPF**. Resuspended cells were lysed by sonication. Lysate was cleared by centrifugation at 30,000 × g and 20 °C for 45 min and supernatant loaded onto a 5 mL HisTrap column (GE Healthcare) equilibrated in lysis buffer. All purification steps were done at room temperature. The HisTrap column was washed in 20 column volumes (CV) buffer W (50 mM Hepes/KOH pH 7.5, 1 M KCl, 50 mM imidazole, 10% v/v glycerol, 1 mM DTT) and 2 CV lysis buffer. Bound protein was eluted in buffer E (50 mM Hepes/KOH pH 7.5, 200 mM KCl, 300 mM imidazole, 10% v/v glycerol, 1 mM DTT) with a 20 CV linear gradient from 50 mM to 300 mM imidazole. Fractions containing *Tt*HPF protein were pooled and 1:100 w/w TEV-protease (made in-house) added. The solution was dialyzed overnight against 2 L buffer D (20 mM Hepes/KOH pH 7.5, 100 mM KCl, 10% v/v glycerol, 1 mM DTT). The dialyzed protein solution was diluted with one volume buffer Q (50 mM Hepes/KOH pH 8.0, 10% v/v glycerol, 1 mM DTT) and loaded on a 9 mL Source15Q column (GE Healthcare) equilibrated in buffer Q50 (same as buffer Q plus 50 mM KCl). The column was washed for 4 CV with buffer Q50 and bound protein eluted with a 20 CV linear gradient to buffer Q1000 (same as buffer S plus 1 M KCl). Peak fractions containing *Tt*HPF were concentrated to ∼ 5 mL volume by centrifugation in VivaSpin concentrators with a 5000 Da cutoff. The protein solution was loaded on a HiLoad Superdex 75 column (GE Healthcare) equilibrated in buffer F (20 mM Hepes/KOH pH 7.5, 100 mM KCl, 10% v/v glycerol, 1 mM DTT). Peak fractions were concentrated to 8 mg/mL. The final yield starting from 45 g cells was 30 mg. Aliquots were flash frozen in liquid nitrogen prior to storage at −80 °C. *Tt*HPF concentration was estimated from absorption in spectrophotometer at 280 nm using theoretical extinction coefficient from ExPASy-ProtParam. For high-resolution gel filtration analysis, purified *Tt*HPF protein was loaded onto a Superdex200 Increase 10/300 column (GE Healthcare) equilibrated in buffer F.

**Isolation of *T. thermophilus* ribosome**. *T. thermophilus* cell pellet was purchased from BFF (Athens Georgia, USA) and ribosomes were isolated based on modified protocols previously published[28,41]. All steps were carried out at 4 °C or on ice. In brief, cells were resuspended in buffer A (20 mM Hepes/KOH pH 7.5, 100 mM NH₄Cl, 10 mM Mg(OAc)₂, 0.5 mM EDTA, 1 mM DTT, 0.1 mM benzamidine) and lysed by high-pressure homogenization with a backpressure of 20,000 psi. This was repeated three times. The lysate was treated with 1 U/g cells RNase-free DNaseI (ThermoFisher) before clearing by centrifugation for 45 min at 30,000 × g. Crude ribosomes were pelleted through sucrose cushion (20 mM Hepes/KOH pH 7.5, 500 mM KCl, 10 mM Mg(OAc)₂, 0.5 mM EDTA, 37% w/v sucrose, 1 mM DTT) by ultracentrifugation for 17 h at 125,171 × g. Ribosomal pellets were briefly washed in buffer C (20 mM Hepes/KOH pH 7.5, 400 mM KCl, 10 mM Mg(OAc)₂, 1.5 M (NH₄)₂SO₄, 1 mM DTT) before resuspended in 5 mL buffer C per pellet. The ribosome solution was loaded on butyl-ToyoPearl column (TOSOH) equilibrated in buffer C. The column was washed for 1 CV with 50% buffer C and 50% buffer D (same as buffer C without (NH₄)₂SO₄) and bound ribosomes were eluted with a 20 CV linear gradient into buffer D. Fractions containing ribosome were pooled and ribosomes pelleted by ultracentrifugation for 17 h at 125,171 × g. Ribosomal pellets were resuspended in buffer RE (10 mM Hepes/KOH pH 7.5, 50 mM KCl, 10 mM NH₄Cl, 10 mM Mg(OAc)₂, 0.25 mM EDTA, 1 mM DTT). Using SW28 tubes, linear 5–20% w/v sucrose gradients in buffer RE were prepared with Gradient Master (Biocomp). Resuspended ribosomes were placed on sucrose gradients and sedimented by ultracentrifugation for 17 h at 31,383 × g. Gradients were fractionated bottom-to-top using peristaltic pump with a connected UV monitor and chart recorder. Fractions of the peak corresponding to 70S ribosomes were pooled, diluted with buffer RE to fit one Ti45 centrifugation tube and ribosomes pelleted by ultracentrifugation for 17 h at 125,171 × g. The resulting ribosome pellet was dissolved in buffer G (5 mM Hepes/KOH pH 7.5, 50 mM KCl, 10 mM Mg(OAc)₂, 10 mM NH₄Cl, 1 mM DTT) to final concentration of 20 mg/mL. Aliquots were flash frozen in liquid nitrogen and stored at −80 °C. Ribosome concentration was estimated from spectrophotometric absorption at 260 nm using an extinction coefficient of 15.0 per 1 mg/mL.

**Analytical ultracentrifugation**. Samples were handled at room temperature and sedimentation velocity experiment was done at 20 °C. Purified 70S ribosome was diluted in buffer G to a final concentration of 1 $A_{260}$ (equal to 0.0667 mg/mL or 30 nM) and mixed with purified *Tt*HPF in a series of molar ratios. The experiments were run in a ProteomeLab-XL-I centrifuge using an An-50Ti rotor spinning at 12,000 rpm with absorbance monitoring at 260 nm scanning every 4 min. Density of sample solution and viscosity of buffer G were calculated using SEDNTERP[42] software and sedimentation data analyzed using SEDFIT[43] assuming a continuous distribution models, c (s). For calculations, a partial specific volume ($\bar{v}$) of 0.64 was used[44].

**Negative stain EM analysis**. In-house carbon coated copper grids (Gilder grids G400-C3) were glow discharged for 45 s at 25 mA (PELCO easiGlow) prior to sample application. Control sample with 70S ribosomes as well as sample with ribosome mixed with *Tt*HPF was applied to grids for one minute using a

concentration of 50 nM and 3.5 uL volume per grid. Sample liquid was blotted off with filter paper and grid stained with uranyl formate (2% w/v). Micrographs were collected on a Tecnai G2 Spirit microscope (FEI) at 120 kV acceleration voltage and 52,000 times magnification, equipped with a 4k×4k CMOS camera (TVIPS).

**Cryo-EM grid preparation**. Isolated 70S ribosomes were mixed in equimolar ratio with purified HPF and incubated for 30 min at 55 °C. The 100S ribosomes were separated from 70S ribosomes by ultracentrifugation in SW41 for 16 h at 16,000 rpm through 5–30% w/v linear sucrose gradients prepared in buffer RE (see above). Fractions of the peak corresponding to 100S were pooled and then concentrated on Amicon Ultra-15 (GE Healthcare) followed by two subsequent rounds of dilution with buffer RE and concentration to remove sucrose that might otherwise perturb the contrast in the micrographs. For cryo-EM grid preparation the ribosome solution was adjusted to 100 nM (0.5 mg/mL) 100S concentration. Quantifoil R1.2/1.3 300 mesh grids were glow discharged for 40 s at 3 mA before use and flash frozen in liquid ethane cooled by liquid nitrogen using a FEI Vitrobot MarkIV.

Similar 100S ribosome sample preparation procedure was used prior to sample application onto Quantifoil R2/2 300 mesh grids with amorphous carbon coating. Sample concentration was adjusted to 13 nM (0.06 mg/mL) with the same Vitrobot settings as described above.

**Cryo-EM data collection and processing**. Data was collected automatically using SerialEM[45] on a spherical aberration ($C_s$) corrected FEI Titan Krios transmission electron microscope at liquid nitrogen temperature operating at an accelerating voltage of 300 kV equipped with a K2 Summit camera (Gatan) at a nominal magnification of 59,000× resulting in a calibrated pixel size of 1.1 Å/pixel. Camera operated in super-resolution mode. Dose per frame was 1.06 e⁻/Å² with 33 frames per acquisition. All image processing steps with movie frame alignment and determination of CTF parameters as well as singe particle processing and refinement was done within cisTEM[46]. Particles from the unsupported ice data set were picked using reference-free automatic picking with manual inspection of all picked positions. Particles were extracted with a 700-pixel box and all 93,133 particles 2D classified. Particles in class averages showing clear structural features for both ribosomes within the dimer were used further in 3D classification and refinement steps. Particles in 3D class IV (see Supplementary Figure 3) were aligned and C2 symmetry was used for the final refinement resulting in a reconstruction at 4.57 Å (0.143 FSC) average resolution. This reconstruction is referred to as 100S (ice). Particles from 3D class II were refined using a 70S mask resulting in a 70S ribosome reconstruction at 3.28 Å (0.143 FSC) average resolution (referred to as 70S (ice)). A similar processing procedure was used for data collected from amorphous carbon grids. After manual inspection of automatically picked particle positions, 55979 particles were used in 2D classification. Following 3D classification and refinement, particles in 3D class II were aligned and C2 symmetry was imposed (see Supplementary Figure 4) in the final refinement resulting in a reconstruction at 4.13 Å (0.143 FSC) average resolution (referred to as 100S (amc)). For all three final reconstructions local resolutions were calculated using ResMap[47].

**Model refinement and validation**. For initial model fitting and building, the 70S ribosome EM density map was sharpened automatically using phenix.auto_sharpen. The crystal structure of the *T. thermophilus* 70S ribosome (PDB entry 4V9B)[48] was edited removing tRNAs, mRNA and antibiotic compound and used as the initial model. The small and large subunit were fitted separately into corresponding density using rigid body fitting in UCSF Chimera[49]. To improve model fit to density, each subunit was further fitted by molecular dynamics flexible fitting with manual inspection and building of the fitted models using Coot[50]. *Tt*HPF-NTD residues 2–122 were built in the density map of the 70S (ice) ribosome guided by a crystallographic model of *Tt*HPF-NTD (PDB entry 2YWQ, unpublished) and de novo building for the linker region. The fitted model of *Tt*70S ribosome with *Tt*HPF-NTD was further used to generate models of *Tt*100S ribosome in 100S (ice) and 100S (amc) EM densities using Namdinator. For *Tt*HPF-CTD building, the *Sa*HPF-CTD (PDB entry 6FXC) was docked into the density and used as a template for model building of residues 129–185. Models of 70S (ice), 100S (ice), and 100S (amc) were all refined using phenix.real_space_refine[51,52]. Finally, refined models were all validated with MolProbity[53] with all statistics listed in Supplementary Table 1. Figures were prepared using UCSF Chimera and PyMol (Schrödinger). Ribosomal proteins are all named based on the new system[54].

## Data availability

Atomic models have been deposited with the Protein Databank with the following accession codes: 70S (ice) PDB accession code 6GZQ, 100S (amc) PDB accession code 6GZZ, and 100S (ice) PDB accession code 6GZX. EM density maps have been deposited within the EMDB with accession codes for 70S (ice) EMD-0101, 100S (ice) EMD-0104, and 100S (amc) EMD-0105. Other data are available from the corresponding author upon reasonable request.

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

## Acknowledgements

The authors acknowledge the support and the use of resources of the French Infrastructure for Integrated Structural Biology FRISBI ANR-10-INBS-05 and of Instruct-ERIC. Specifically we would like to thank Catherine Birck for help with the analytical ultracentrifugation experiments, Corrine Crucifix for help with EM sample preparation and verification and Julio Ortiz for help with cryo EM data collection. Special thanks also goes to Rune Kidmose for valuable help in initial stages of model building.

## Author contributions

R.K.F., N.B. and L.B.J. performed the biochemical experiments; R.F.K. and L.B.J. collected the EM data; R.K.F. did data processing, model building and refinement; R.K.F. and L.B.J. analysed the models; M.Y. and L.B.J. provided resources; L.B.J. supervised the project; R.

F.K, N.B., M.Y., and L.B.J. discussed results; R.K.F. wrote the initial draft. R.K.F and L.B.J. edited the manuscript.

## Additional information

**Competing interests:** The authors declare no competing interests.

