## [Peer Review File · Nature Communications]

Reviewers' comments:

Reviewer #1 (Remarks to the Author):

In many bacteria, formation of hibernating 100S ribosomes results from the binding of the long-form hibernation promotion factor (HPF). The N-terminal domain (NTD) of HPF is homologous to pY and short-form HPF, which bind to the small subunit and overlap the A- and P-sites. Last year, four cryo-EM structures confirmed that the NTD binding site but more interestingly showed that dimerization of 70S to build 100S occurs because of dimerization of the C-terminal domain (CTD), which is located at the platform of the small subunit, leading to a unique dimerization arrangement unlike that observed for *E. coli* 100S.

The manuscript of Flygaard et al presents a cryo-EM structure of *Thermus thermophilus* 100S ribosome with an average resolution of 4-5Å, as well as a 70S ribosome from the hibernating 100S dimer at 3.2Å. Most results are consistent with previous studies and relatively little new information is gained, suggesting in contrast to the title of the paper, this study sheds very little new light on the hibernation mechanism. If anything, it supports and confirms the previous studies and contributes to the notion of some minor species-specific differences in the dimerization conformation/arrangement. Importantly, it remains unclear why the authors formed the 100S in vitro rather than looking at in vivo formed 100S particles? How can the authors be sure that the in vitro formed 100S will be the same as the in vivo isolated particles? Last, the advantage of having established the in vitro 100S formation assay would be then to interrogate using mutagenesis critical residues of HPF, however, this appears not to have been done.

Some more points:

1. The authors show that TtHPF is most likely a dimer in solution (Figure 1), consistent with previous studies. The authors make no additional analysis using mutagenesis for example to dissect the dimer interface, therefore, this data could be supplemental since it is confirmatory.
2. The cryo-EM structure of the 100S is reported with a slightly better average resolution than the best of the previous studies, however, since no local resolution images are shown for the ligands, whether there is an improvement in the critical/interesting areas cannot be assessed. This information should be added for Supplementary Fig. 4.
3. Regardless of the local resolution of the 100S, the additional 3.2Å reconstruction of the 70S from the 100S allows a better description of the contacts of the NTD with the 30S. The NTD is homologous to the pY and short-form HPF, which were already described at high resolution on the ribosome, therefore, this represents a relatively modest increase in new information. Moreover, the Sup Fig. 6 images do not inspire confidence in the resolution and modeling of the interactions – R18 in Panel B appears to have density extending up from the ribbon as well. R9 in panel C also appears not to be well-positioned in the density? R27 in panel D does not appear to have density yet is modelled and P112 does not appear to be optimally positioned for stacking with Y25 in panel F? Some supporting mutagenesis experiments might be useful to validate the importance/existence of interactions?
4. Compared to the previous 100S reconstructions, the authors observe extra density for additional residues of the linker that spans between the NTD and the CTD. Unfortunately, this appears only to be in the 70S and not in the 100S reconstruction, raising the question of whether the conformation is the same for in vitro 100S, let alone in vivo 100S particles. Moreover, the authors make no attempt to biochemically validate the importance of the linker in 100S formation, as was previously performed for other long-form HPFs.
5. The CTD conformation appears to be similar to that as observed in the previous 100S reconstructions, however, the local resolution appears to be lower than the average resolution so its unclear whether there is any gain in information in this region? Nevertheless, the dimer interface of the Tt100S differs somewhat from previous 100S reconstructions due mainly to the shortened h26. Differences were already evident when comparing other 100S reconstructions, however, here the dimer interface does not involve ribosome-ribosome interactions, only interactions via the CTDs. Again no attempts are made to validate important interactions between

CTDs or CTDs with the ribosome.

Reviewer #2 (Remarks to the Author):

In this manuscript entitled "Cryo-EM structure of the *Thermus thermophilus* 100S ribosome sheds new light on hibernation mechanism", the authors address the molecular mechanism of ribosome dimer formation in *T. thermophilus*. They do so by using cryo-EM and single particle analysis to determine the 3D structure of a complex of two 70S ribosomes forming a dimer through interactions with the dimeric hibernation promoting factor (HPF) protein. While the study is based primarily on structural analysis, the authors also include some biochemical data used to determine how differences in 70S ribosome and HPF stoichiometries affects Tt100S dimer formation. Overall the paper is well presented and the structures present some differences of HPF interactions for *T. thermophilus* versus other bacterial 100S ribosome dimers that have previously been published. The primary critique lies with the level of new information provided from the present study as compared to the previously published 100S dimer structures. More detailed comments are included below:

The authors present a very thorough interpretation, but the primary critique remains that the structure is similar to previous structures, albeit from other bacteria, and therefore does not shed significant new insight into 100S formation and function over what was previously known.

One of the key contributions of the paper (claimed by authors) is that the linker connecting NTD and CTD of TtHPF can be resolved in the 100S structure. However, this statement is difficult to defend as the only density attributed to this region is very noisy (Fig. 3A).

The figures (Fig. 3) illustrate the positioning of TtHPF in relation to tRNA and mRNA molecules. The authors should clearly state that the structure does not contain mRNA and/or tRNA, but that they are simply modeled in based on previously solved structures. The bigger question, however, is that it remains unclear whether TtHPF even binds with mRNA present. If so, the authors may not be able to distinguish between densities assigned to mRNA versus TtHPF. If not, then the argument that TtHPF binding impairs Shine-Dalgarno and anti-Shine-Dalgarno interactions would be invalid because there would never be any Shine-Dalgarno sequence due to TtHPF binding.

A similar point that remains unclear is at what point does TtHPF (or any other HPF) bind to the ribosomes to impede translation. It is unclear how binding to vacant 70S ribosomes occurs when the ribosomes are required to exist as individual subunits (30S and 50S) to initiate translation. Also, as TtHPF does not appear to be able to bind to translating ribosomes (due to clashes with mRNA-tRNA) how does it function to impair translation? At what point of the translation cycle does TtHPF bind?

As evidenced by the two separate 100S structures determined in this report (100S ice and 100S amc) the slightly different conformations can be attributed to the use of substrate support on the cryo-EM grids. Therefore, there is a concern that similar differences in sample prep might explain the differences in conformation for Tt100S vs Bs, Ec, or Sa100S particles that are discussed in Fig. 5.

In general, the points being made in the figures are not always clear; perhaps due to illustrative challenges. As an example, Figure 2 is not as informative as what is illustrated in Supplemental Figure 1 (SF1A). Perhaps the 2 figures could be merged to make the points clearer.

Similarly, in (Figure 2B and D; middle page 7), the authors state that the TtHPF-NTD was observed, but that the TtHPF-CTD is not observed. It is nearly impossible for the reader to judge the accuracy of these statements by looking at the figures that are presented. Similar for Figure

2C (and many other figures).

The figures (and supplemental figures) are cited in random, instead of consecutive, order.

It is not at all intuitive that, in SF2A, a TtHPF monomer is 21 kDa, but the 'apparent' weight runs at 76 kDa and on a denaturing gel there is a major band at 45 kDa that the authors attribute to being TtHPF protein that is not fully denatured?

Similar to above, it is not clear why the authors interpreted the gel-filtration elution volume of TtHPF as a homodimer in an extended conformation (Fig. 2A). There is not sufficient evidence to rule out other possibilities, such as a trimer (or higher oligomeric state) in a compact state.

The relative terms such as "in general was not very good" and "70S (ice) reconstruction was of very high quality" top of pg. 9 should be rephrased to better make the points being made.

Reviewer #3 (Remarks to the Author):

In the presented study by Flygaard et al., the authors obtained high-resolution Cryo-EM reconstruction of the 100S ribosome dimer from the Gram-negative bacterium *Thermus thermophilus* that, unlike *E. coli*, has a long version of the Hibernation Promoting Factor. The process of ribosome inactivation through 100S formation is referred in the literature as ribosome hibernation and plays a pivotal role in the survival of bacteria during harsh conditions including treatment with antibiotics and nutrient starvation. Studying of this process in various model microorganisms attracted lots of interest recently because there are some indications that in addition to survival this process can lead to the development of drug resistance.

After reading the abstract of this manuscript, honestly, I was not much excited and was expecting yet another structure of the 100S ribosome particle from yet another microorganism. Four such structures have been already published last year by four independent groups (including one by the Yusupov group). However, I was pleasantly surprised and excited after reading the main text. I think the authors communicated several very important findings that definitely merit publication in *Nature Communications*.

Among the strong sides of the current work, I would like to point out the following results:

1. This work presents the highest resolution reconstruction of the complete! 100S particle that was reported to date. Previously, structures of 100S dimers were reported at 5.6Å from *Lactococcus lactis* (by Guskov group) and at 6.76Å from *S. aureus* (by Yonath Group). In the two more structures of the 100S dimers from *S. aureus* and *B. subtilis* by Yusupov and Wilson groups the resolution of 3.7Å and 3.9Å was mainly attributed to the 70S monomer ribosome within the 100S dimer, not the entire 100S dimer. Therefore, by far the current structure is the most detailed visualization of the 100S ribosome;
2. Compared to the previous structures of the 100S dimers, in which the linker connecting the NTD and CTD of the LHPF protein was distorted, in this work authors managed to obtain a decent quality electron density map for the corresponding region. Although few amino acids are still missing from the density, this reconstruction is by far more complete than the previous ones;
3. The finding that the 70S ribosomes most efficiently dimerize into 100S dimers only at the equimolar ratio of LHPF to 70S ribosome is interesting by itself because one would expect a sigmoidal shape of this dependency with a plateau at molar ratios higher than 1:1;
4. The current structure of the 100S dimer from another organism yet again revealed species-

specific differences of the dimer organization, such as the absence of ribosome-to ribosome contacts. This is interesting because hibernation factors from different bacteria are homologous, and ribosomes from different species are also conceptually the same, however, the 100S dimers from these species appear to be different.

In summary, the authors shared an interesting story about the organization of the 100S ribosome dimers in *Thermus thermophilus* – the beloved source of ribosomes for crystallographic studies of the ribosome. In my opinion, this manuscript is very well written with very good and clear illustrations that are self-explanatory. It was actually fun to read, especially the discussion section. This work provides important results and shall be published. This reviewer also has several minor critical points, which the authors might wish to address:

Comments, suggestions and questions to the authors:

1. One of the novel findings in this work is that excess of LHPF inhibits 100S dimer formation. However, the explanation provided by the authors in the text is totally unclear. The authors attribute this effect to the strong binding of the TtLHPF to the ribosome, however, it is unclear how that will lead to the decreased dimer formation. I would suggest elaborating more on this explanation in both the “Results” and “Discussion” sections. Also, it would be great if authors could logically connect this explanation with the points that they bring up in the discussion, such as (i) TtLHPF may not be fully active, (ii) not all 70S ribosomes are converted to 100S dimers, which concurs with the idea that a certain amount of 70S is still needed during stress.
2. The last sentence of the “Results” section might be toned down a bit by including the name of the microorganism “*Thermus thermophilus*”. Otherwise, it sounds like there are no ribosome-to-ribosome contacts in any of the previously studied dimers, which we know is not the case.
3. The authors noted that the linker between NTD and CTD of the TtLHPF protein overlaps with the ASD/SD region and can sterically clash with potential ASD/SD helix, similar to RMF in *E.coli*. It doesn't seem to me that this is the actual role of the linker, because unlike RMF in *E.coli* this linker is not a separate protein but rather a part of a longer polypeptide. In my opinion, it is just a coincidence that the linker is located in the same region as RMF. The role of the linker might be to ensure that the dimerization domain (CTD) is delivered to the ribosome at the same time as the NTD blocks binding of the mRNA and all tRNAs.
4. In Figure 3, panels F and G are somewhat redundant, one of them could be removed.
5. In Figure 3, panel H could be made larger. Currently, it is hard to see the elements, especially in the close-up view.
6. The authors might wish to revise their abstract so that their most exciting findings are adequately communicated to the potential reader.

→ We wish to thank all reviewers for many helpful comments and suggestions. Below are our point by point comments / changes.

In many bacteria, formation of hibernating 100S ribosomes results from the binding of the long-form hibernation promotion factor (HPF). The N-terminal domain (NTD) of HPF is homologous to pY and short-form HPF, which bind to the small subunit and overlap the A- and P-sites. Last year, four cryo-EM structures confirmed that the NTD binding site but more interestingly showed that dimerization of 70S to build 100S occurs because of dimerization of the C-terminal domain (CTD), which is located at the platform of the small subunit, leading to a unique dimerization arrangement unlike that observed for E.coli 100S.

The manuscript of Flygaard et al presents a cryo-EM structure of *Thermus thermophilus* 100S ribosome with an average resolution of 4-5Å, as well as a 70S ribosome from the hibernating 100S dimer at 3.2Å. Most results are consistent with previous studies and relatively little new information is gained, suggesting in contrast to the title of the paper, this study sheds very little new light on the hibernation mechanism. If anything, it supports and confirms the previous studies and contributes to the notion of some minor species-specific differences in the dimerization conformation/arrangement.

→ We wish to respectfully disagree with this reviewer's comments that our study contributes only little. Our structures do represent the, to date, highest resolution structures of 100S particles. It is a native homologous complex that can validate or reject suggestions based on previous heterologous ribosome / hibernation factor complexes, and as such very important. Furthermore, the HPF N-terminal domain / ribosome interaction as well as the linker region is resolved to a much higher degree in our structure and described in much higher detail.

Importantly, it remains unclear why the authors formed the 100S *in vitro* rather than looking at *in vivo* formed 100S particles? How can the authors be sure that the *in vitro* formed 100S will be the same as the *in vivo* isolated particles?

→ In previous studies no difference was found between *in vitro* staphylococcus 100S particles (Khusainov 2017 (Yusupov lab)) and *in vivo* formed staphylococcus 100S particles (Matzov 2017 (Yonath lab)). So, we decided to use the *in vitro* approach to be able to achieve higher homogeneity and to better control the experiment.

Last, the advantage of having established the *in vitro* 100S formation assay would be then to interrogate using mutagenesis critical residues of HPF, however, this appears not to have been done.

→ It has already been established that the C-terminal domain is sufficient for 100S dimerization (Matzov 2017 (Yonath lab)). The C-terminal in our study (and in all other 100S studies) is only resolved to low resolution 4.5Å. It is higher than all previous structures but still not high enough to suggest critical residues in this part of the structure. Our high-resolution structure is of the N-terminal domain and that is most likely involved in translational arrest and therefore our 100S formation assay (AUC) will not give any useful information using mutagenesis of the side chains of the N-terminal domain. Another type of assay should be used for this, a translational assay, but we find that that is beyond the scope of this paper and warrants a dedicated study.

Some more points:

1. The authors show that TtHPF is most likely a dimer in solution (Figure 1), consistent with previous studies. The authors make no additional analysis using mutagenesis for example to dissect the dimer interface, therefore, this data could be supplemental since it is confirmatory.

→ No previously published studies have shown what we show in figure 1, that is the dimerization / formation of 100S as a function of *t*7HPF concentration. The profile from the gel filtration experiment where we confirm that HPF in solution is a dimer is already in supplementary material (Supplementary Fig 2).

2. The cryo-EM structure of the 100S is reported with a slightly better average resolution than the best of the previous studies, however, since no local resolution images are shown for the ligands, whether there is an improvement in the critical/interesting areas cannot be assessed. This information should be added for Supplementary Fig. 4.

→ We have created new figures (Supplementary figures 4 B and C) showing the local resolution of the HPF in the different (ice) reconstructions. This clearly demonstrate that the local resolution for the HPF ligand is similar to the overall resolution of the structures and therefore that there is significant improvement, also in these areas, compared to previous structures.

3. Regardless of the local resolution of the 100S, the additional 3.2Ang reconstruction of the 70S from the 100S allows a better description of the contacts of the NTD with the 30S. The NTD is homologous to the pY and short-form HPF, which were already described at high resolution on the ribosome, therefore, this represents a relatively modest increase in new information.

→ The study being referred to here is the 70S *Thermus thermophilus* crystal structure with *E.coli* proteins pY and short-form HPF. This is a heterologous complex from two species that have significantly different modes of dimerization and we therefore feel that our study of the “true” native complex is of much higher value than the previous study. We very much disagree that this is only a modest increase in new information, in principle the heterologous complex might have been totally wrong and our study is therefore very important.

Moreover, the Sup Fig. 6 images do not inspire confidence in the resolution and modeling of the interactions – R18 in Panel B appears to have density extending up from the ribbon as well. R9 in panel C also appears not to be well-positioned in the density? R27 in panel D does not appear to have density yet is modelled and P112 does not appear to be optimally positioned for stacking with Y25 in panel F? Some supporting mutagenesis experiments might be useful to validate the importance/existence of interactions?

→ We have changed the figures in supplementary figure 6 panels B, C, D and F to better show the sidechain densities and inspire confidence. R27 is out of density at this contour level but if we contour lower to see the density the rest becomes very difficult to distinguish. We do feel that this is entirely normal and accepted in the structural community. Again, we feel that the suggested mutagenesis experiments are beyond the scope of this paper and warrants a dedicated functional study where the influence on translational arrest is studied to gain any information from such mutagenesis studies.

4. Compared to the previous 100S reconstructions, the authors observe extra density for additional residues of the linker that spans between the NTD and the CTD. Unfortunately, this appears only to be in the 70S and not in the 100S reconstruction, raising the question of whether the conformation is the same for in vitro 100S, let alone in vivo 100S particles. Moreover, the authors make no attempt to biochemically validate the importance of the linker in 100S formation, as was previously performed for other long-form HPFs.

→ The 70S reconstruction was created using only 100S particles (only 100S particles were picked) and therefore the HPF is identical in the two. We have added an extra figure to the supplements showing a superposition of the HPFs (Supplementary 5C). As the reviewer says, truncation of the linker region has

already been done where increasing deletion of the linker region results in a moderate decrease in 100S formation (Beckert 2017 (Wilson lab)).

In the manuscript we have added the following:

“As the conformation of TtHPF was found to be identical in the three different reconstructions (Supplementary Figure 5C), we initially thought of combining 100S particles from the two data sets (ice and amc) aiming for a higher resolved reconstruction “

5. The CTD conformation appears to be similar to that as observed in the previous 100S reconstructions, however, the local resolution appears to be lower than the average resolution so its unclear whether there is any gain in information in this region?

→ We have added Supplemental figure 5 panel B to show local resolution for the HPF in the 100S carbon (amc) reconstruction including the CTD. This figure shows that the local resolution is 4 – 5Å and therefore significantly better than previous structures.

Nevertheless, the dimer interface of the Tt100S differs somewhat from previous 100S reconstructions due mainly to the shortened h26. Differences were already evident when comparing other 100S reconstructions, however, here the dimer interface does not involve ribosome-ribosome interactions, only interactions via the CTDs. Again no attempts are made to validate important interactions between CTDs or CTDs with the ribosome.

→ The novelty in our study is the comparison between our structure and other species having long HPF. It has been proposed before that due to a shortening of h26 in some of these species there might be a difference in dimerization (Khusanov 2017) but an actual structure of this new dimerization mode has never existed before now. We have not proposed any specific interactions between the CTDs in the dimerization interface because the resolution is too low (4.5Å) to structurally pinpoint specific residues as being critical for this interaction.

Reviewer #2 (Remarks to the Author):

In this manuscript entitled "Cryo-EM structure of the *Thermus thermophilus* 100S ribosome sheds new light on hibernation mechanism", the authors address the molecular mechanism of ribosome dimer formation in *T. thermophilus*. They do so by using cryo-EM and single particle analysis to determine the 3D structure of a complex of two 70S ribosomes forming a dimer through interactions with the dimeric hibernation promoting factor (HPF) protein. While the study is based primarily on structural analysis, the authors also include some biochemical data used to determine how differences in 70S ribosome and HPF stoichiometries affects Tt100S dimer formation. Overall the paper is well presented and the structures present some differences of HPF interactions for *T. thermophilus* versus other bacterial 100S ribosome dimers that have previously been published. The primary critique lies with the level of new information provided from the present study as compared to the previously published 100S dimer structures. More detailed comments are included below:

The authors present a very thorough interpretation, but the primary critique remains that the structure is similar to previous structures, albeit from other bacteria, and therefore does not shed significant new insight into 100S formation and function over what was previously known.

One of the key contributions of the paper (claimed by authors) is that the linker connecting NTD and CTD of TtHPF can be resolved in the 100S structure. However, this statement is difficult to defend as the only density attributed to this region is very noisy (Fig. 3A).

→ We have added a new panel to figure 3 showing the density of the linker region. This figure clearly shows the high quality of the linker and should inspire confidence with the reader.

The figures (Fig. 3) illustrate the positioning of TtHPF in relation to tRNA and mRNA molecules. The authors should clearly state that the structure does not contain mRNA and/or tRNA, but that they are simply modeled in based on previously solved structures.

→ We agree with the reviewer and have changed the figure 3E to remove the surface representation for mRNA and tRNAs to not confuse the reader into thinking that we have tRNAs and mRNAs in our structure. Similar for Figure 3H.

The bigger question, however, is that it remains unclear whether TtHPF even binds with mRNA present. If so, the authors may not be able to distinguish between densities assigned to mRNA versus TtHPF. If not, then the argument that TtHPF binding impairs Shine-Dalgarno and anti-Shine-Dalgarno interactions would be invalid because there would never be any Shine-Dalgarno sequence due to TtHPF binding.

A similar point that remains unclear is at what point does TtHPF (or any other HPF) bind to the ribosomes to impede translation. It is unclear how binding to vacant 70S ribosomes occurs when the ribosomes are required to exist as individual subunits (30S and 50S) to initiate translation. Also, as TtHPF does not appear to be able to bind to translating ribosomes (due to clashes with mRNA-tRNA) how does it function to impair translation? At what point of the translation cycle does TtHPF bind?

→ No studies have been done to show how HPF enters the ribosomes and whether it is able to bind ribosomes containing tRNAs and mRNA and this is therefore unknown. The goal of our experiment was not to answer this question because a structural study would not suffice to elucidate this mechanism. It would require biochemical and functional studies rather than structural studies. However, all structural studies done on *Thermus thermophilus* ribosomes isolated using similar purification protocols have shown that they do not contain mRNA nor A- and P-tRNA. We are therefore 100% sure that the density we see for HPF cannot be mRNA since we form the 100S in vitro without adding mRNA and therefore our argument is valid. We don't feel that we should add any explanatory sentences to the manuscript because it is not relevant to our observations and it would confuse the readers even more.

As evidenced by the two separate 100S structures determined in this report (100S ice and 100S amc) the slightly different conformations can be attributed to the use of substrate support on the cryo-EM grids. Therefore, there is a concern that similar differences in sample prep might explain the differences in conformation for Tt100S vs Bs, Ec, or Sa100S particles that are discussed in Fig. 5.

→ We have added supplementary figure 7 panel E to show a superposition of the 100S (ice) and 100S (amc). This figure shows that the conformation is identical.

We have added the following figure text:

Orthogonal views showing superposition of Tt100S (ice) and Tt100S (amc) reconstructions (grey and gold coloured, respectively) with the EM reconstructions of *B. subtilis* 100S (EMD-3664, green) and *S. aureus* 100S (EMD-3637, purple and EMD-3638, salmon). This figure demonstrates that the overall conformation of the 100S is identical although there are subtle differences in the dimerization interface.

And we have changed the sentence in the text to (page 10):

However, although the overall conformation of the 100S (ice) and 100S (amc) was identical, there was a slight difference when looking specifically at the dimerization interface where the 100S (amc) has two additional sites of interaction between the small subunit head domains centered at ribosomal proteins uS7 and uS9 (Supplementary Figure 7A).

In general, the points being made in the figures are not always clear; perhaps due to illustrative challenges. As an example, Figure 2 is not as informative as what is illustrated in Supplemental Figure 1 (SF1A). Perhaps the 2 figures could be merged to make the points clearer.

Similarly, in (Figure 2B and D; middle page 7), the authors state that the TtHPF-NTD was observed, but that the TtHPF-CTD is not observed. It is nearly impossible for the reader to judge the accuracy of these statements by looking at the figures that are presented. Similar for Figure 2C (and many other figures).

→ Supplementary figure 1 depicts previous structures for comparison so we feel that it is more appropriate for the supplementary material. We have changed figure 2D to clearly show the density for the NTD as well as the missing density for the CTD for the 70S(ice) reconstruction. We changed the figure legend to: D) View of 70S (ice) with TtHPF-NTD colored in orange. Close-up views on 30S subunit show location of TtHPF-NTD and the linker region. There was no density for the TtHPF-CTD in 70S (ice) reconstruction.

The figures (and supplemental figures) are cited in random, instead of consecutive, order.

→ We have corrected this in the manuscript.

It is not at all intuitive that, in SF2A, a TtHPF monomer is 21 kDa, but the 'apparent' weight runs at 76 kDa and on a denaturing gel there is a major band at 45 kDa that the authors attribute to being TtHPF protein that is not fully denatured?

Similar to above, it is not clear why the authors interpreted the gel-filtration elution volume of TtHPF as a homodimer in an extended conformation (Fig. 2A). There is not sufficient evidence to rule out other possibilities, such as a trimer (or higher oligomeric state) in a compact state.

→ It is already written in the main text that the discrepancy is due to the shape of the HPF dimer but we have inserted the following in the figure text for Supplementary figure 2:

These results indicate that TtHPF most likely is a dimer in solution although we cannot rule out other stoichiometries. Probably due to its elongated shape it elutes as a larger protein than the 45 kDa that is expected. Given the nature of HPF, a trimer would be very unlikely.

The relative terms such as "in general was not very good" and "70S (ice) reconstruction was of very high quality" top of pg. 9 should be rephrased to better make the points being made.

→ We have rephrased the main text to:

In previous structures of long HPF proteins, the density for the LHPF-NTD and the linker region was poorly resolved allowing the linker region to only be traced to approximately residue 101 to 106 depending on the study²³⁻²⁵. In our 3.28 Å 70S (ice) reconstruction the electron density for both the NTD and the linker was well resolved (Figure 2D and 3A+B) and allowed us to build the TtHPF linker until residue 122, only leaving a gap of 7 residues between the N and C-terminal domains (Figure 3F).

Reviewer #3 (Remarks to the Author):

In the presented study by Flygaard et al., the authors obtained high-resolution Cryo-EM reconstruction of the 100S ribosome dimer from the Gram-negative bacterium *Thermus thermophilus* that, unlike *E. coli*, has a long version of the Hibernation Promoting Factor. The process of ribosome inactivation through 100S formation is referred in the literature as ribosome hibernation and plays a pivotal role in the survival of bacteria during harsh conditions including treatment with antibiotics and nutrient starvation. Studying of this process in various model microorganisms attracted lots of interest recently because there are some indications that in addition to survival this process can lead to the development of drug resistance.

After reading the abstract of this manuscript, honestly, I was not much excited and was expecting yet another structure of the 100S ribosome particle from yet another microorganism. Four such structures have been already published last year by four independent groups (including one by the Yusupov group). However, I was pleasantly surprised and excited after reading the main text. I think the authors communicated several very important findings that definitely merit publication in *Nature Communications*.

Among the strong sides of the current work, I would like to point out the following results:

1. This work presents the highest resolution reconstruction of the complete! 100S particle that was reported to date. Previously, structures of 100S dimers were reported at 5.6Å from *Lactococcus lactis* (by Guskov group) and at 6.76Å from *S. aureus* (by Yonath Group). In the two more structures of the 100S dimers from *S. aureus* and *B. subtilis* by Yusupov and Wilson groups the resolution of 3.7Å and 3.9Å was mainly attributed to the 70S monomer ribosome within the 100S dimer, not the entire 100S dimer. Therefore, by far the current structure is the most detailed visualization of the 100S ribosome;
2. Compared to the previous structures of the 100S dimers, in which the linker connecting the NTD and CTD of the LHPF protein was distorted, in this work authors managed to obtain a decent quality electron density map for the corresponding region. Although few amino acids are still missing from the density, this reconstruction is by far more complete than the previous ones;
3. The finding that the 70S ribosomes most efficiently dimerize into 100S dimers only at the equimolar ratio of LHPF to 70S ribosome is interesting by itself because one would expect a sigmoidal shape of this dependency with a plateau at molar ratios higher than 1:1;
4. The current structure of the 100S dimer from another organism yet again revealed species-specific differences of the dimer organization, such as the absence of ribosome-to-ribosome contacts. This is interesting because hibernation factors from different bacteria are homologous, and ribosomes from different species are also conceptually the same, however, the 100S dimers from these species appear to be different.

In summary, the authors shared an interesting story about the organization of the 100S ribosome dimers in *Thermus thermophilus* – the beloved source of ribosomes for crystallographic studies of the ribosome. In my opinion, this manuscript is very well written with very good and clear illustrations that are self-explanatory. It was actually fun to read, especially the discussion section. This work provides important results and shall be published. This reviewer also has several minor critical points, which the authors might wish to address:

Comments, suggestions and questions to the authors:

1. One of the novel findings in this work is that excess of LHPF inhibits 100S dimer formation. However, the explanation provided by the authors in the text is totally unclear. The authors attribute this effect to the strong binding of the TtLHPF to the ribosome, however, it is unclear how that will lead to the decreased

dimer formation. I would suggest elaborating more on this explanation in both the “Results” and “Discussion” sections. Also, it would be great if authors could logically connect this explanation with the points that they bring up in the discussion, such as (i) TtLHPF may not be fully active, (ii) not all 70S ribosomes are converted to 100S dimers, which concurs with the idea that a certain amount of 70S is still needed during stress.

→

In results we added the following to the text:

This observation could be explained by a strong binding of TtHPF to the Tt70S ribosome, whereby TtHPF binding sites on ribosomes are saturated and thus acting inhibiting to Tt100S ribosome formation (Figure 1C). We did not observe a complete conversion of 70S to 100S dimers which lead us to believe that either TtHPF is not fully active at the temperature where the experiment was performed or perhaps a population of the 70S ribosomes are protected from dimerization in some way that we could not detect.

In the discussion we have added the following to the text:

Despite the clear formation of Tt100S ribosomes indicated by the 100S peaks in the sedimentation profiles, we also observed a 70S sedimentation peak that we naturally attribute to 70S ribosomes. In the experiment with Tt70S ribosome and TtHPF mixed in 1:1 molar ratio, approximately only half of the Tt70s ribosomes are converted to Tt100S ribosomes. Whether this is a reflection of our purified TtHPF protein not being fully active is difficult to assess. It could also be that the temperature of 20°C during the AUC experiment is inhibiting for 100S ribosome formation given the thermophilic nature of the source organism. However, no other experiments of 100S formation from other species has ever reported complete conversion – often the conversion rate has been quite low^{12,14,24}. Perhaps the incomplete conversion to 100S reflects yet undiscovered properties of the mechanism of dimerization or that a certain populace within the ribosome pool are protected from dimerization. This would prevent a complete shutdown of translation and ascertain that some 70S ribosomes are still available for protein production during stress. The decrease in Tt100S ribosome formation upon excess molar ratios of TtHPF we interpret as all possible binding sites on single ribosomes filling up with TtHPF homodimers effectively leaving no vacant Tt70S ribosomes to form 100S dimers (Figure 1C).

2. The last sentence of the “Results” section might be downtoned a bit by including the name of the microorganism “*Thermus thermophilus*”. Otherwise, it sounds like there are no ribosome-to-ribosome contacts in any of the previously studied dimers, which we know is not the case.

→ we have changed the sentence to: These results provide evidence that formation of *Thermus thermophilus* 100S ribosome dimers by long HPF proteins is attributed to the LHPF protein alone and not inter-ribosome interactions.

3. The authors noted that the linker between NTD and CTD of the TtLHPF protein overlaps with the ASD/SD region and can sterically clash with potential ASD/SD helix, similar to RMF in E.coli. It doesn't seem to me that this is the actual role of the linker, because unlike RMF in E.coli this linker is not a separate protein but rather a part of a longer polypeptide. In my opinion, it is just a coincidence that the linker is located in the same region as RMF. The role of the linker might be to ensure that the dimerization domain (CTD) is delivered to the ribosome at the same time as the NTD blocks binding of the mRNA and all tRNAs.

→ We have changed the following in the discussion section: However, the mechanism proposed for RMF in inhibition of translation¹¹ possibly applies for the TtHPF linker region connecting the NTD and CTD. The linker region extends to a region, which would clash with formation of the SD-ASD helix during translation initiation (Figure 3G) providing yet another inhibitory mechanism for TtHPF on translation. Alternatively, the presence of the linker region in the same area as RMF is coincidental and the role of

the linker region may be in synchronization of the dimerization by the CTD and the blocking of mRNA and tRNA binding by the NTD.

4. In Figure 3, panels F and G are somewhat redundant, one of them could be removed.

→ we agree and have removed panel F

5. In Figure 3, panel H could be made larger. Currently, it is hard to see the elements, especially in the close-up view.

→ We agree and have increased the size (Now in panel G). Also, when removing the surface representation of the tRNA and mRNA the figure has been made clearer.

6. The authors might wish to revise their abstract so that their most exciting findings are adequately communicated to the potential reader.

→ We have made the following changes to the abstract:

In response to cellular stresses bacteria conserve energy by dimerization of ribosomes into inactive hibernating 100S ribosome particles. Ribosome dimerization in *Thermus thermophilus* is facilitated by hibernation-promoting factor (TtHPF). In this study we demonstrate high sensitivity of Tt100S formation to the levels of TtHPF and show that a 1:1 ratio leads to optimal dimerization. We report structures of the *T. thermophilus* 100S ribosome determined by cryo-electron microscopy to average resolutions of 4.13 Å and 4.57 Å. In addition, we present a 3.28 Å high-resolution cryo-EM reconstruction of a 70S ribosome from a hibernating ribosome dimer and reveal a role for the linker region connecting the TtHPF N- and C-terminal domains in translation inhibition by preventing Shine-Dalgarno duplex formation. Our work demonstrates that species-specific differences in the dimerization interface govern the overall conformation of the 100S ribosome particle and that for *Thermus thermophilus* no ribosome-ribosome interactions are involved in the interface.